DOI: 10.1038/s41467-018-04728-1　　OPEN

# Cultural conformity generates extremely stable traditions in bird song

Robert F. Lachlan[1], Oliver Ratmann[2] & Stephen Nowicki [3,4]

Cultural traditions have been observed in a wide variety of animal species. It remains unclear, however, what is required for social learning to give rise to stable traditions: what level of precision and what learning strategies are required. We address these questions by fitting models of cultural evolution to learned bird song. We recorded 615 swamp sparrow (*Melospiza georgiana*) song repertoires, and compared syllable frequency distributions to the output of individual-based simulations. We find that syllables are learned with an estimated error rate of 1.85% and with a conformist bias in learning. This bias is consistent with a simple mechanism of overproduction and selective attrition. Finally, we estimate that syllable types could frequently persist for more than 500 years. Our results demonstrate conformist bias in natural animal behaviour and show that this, along with moderately precise learning, may support traditions whose stability rivals those of humans.

[1] Department of Biological and Experimental Psychology, Queen Mary University of London, London E1 4NS, UK. [2] Department of Mathematics, Imperial College London, London SW7 2AZ, UK. [3] Department of Biology, Duke University, Durham, NC 27708, USA. [4] Department of Neurobiology, Duke University Medical School, Durham, NC 27708, USA. Correspondence and requests for materials should be addressed to R.F.L. (email: r.f.lachlan@qmul.ac.uk)

Social learning underlies a wide range of behaviour in many animal species. Yet while cultural transmission is widespread among animals, human culture is unique in its complexity and tendency to accumulate adaptive variations[1,2]. One proposed explanation for this apparent paradox is that cultural transmission in animals relies on mechanisms that are simply not precise enough to support long-lasting traditions shared by large numbers of individuals, and that this prevents the accumulation of complex culture[1,3–6]. It has also been argued that humans are unique in the types of strategies we employ to decide who to learn from, and these strategies may help us maintain stable traditions in the face of innovation and errors in transmission[7–10]. Two central challenges, therefore, are to understand the social learning processes underlying animal traditions, and to assess how stable such traditions may be.

Of various social learning strategies, conformity has been most studied[7,10]. If individuals sample the behavioural variants of a subset of their population, and select variants to learn from at random from within that subset, then they will select variants with probabilities approximating the frequency of those variants in the population. If individuals have a conformist bias, they are more likely to pick a common variant from the sampled set than expected by chance[7]. Conformist biases therefore cause common variants in the population to increase in frequency and select against rare variants. Because new variants are rare, conformist biases tend to remove them, theoretically allowing stable traditions even if innovation rates are high. Often regarded as central to human culture[10], the evidence for conformist biases in animals is growing. This evidence includes experimental and observational evidence that demonstrates animals are paying attention to group norms, and the behaviour of the majority[11–16]. Explicit tests of conformist biases (that is a disproportionate tendency to copy the majority) are rarer, and include only one study in captive sticklebacks[17], and two studies of free-living population of great tits interacting with experimental equipment[18,19]. Together, these studies suggest that animals may frequently employ conformist biases to decide whom to learn from. In this study, we examine whether conformist biases are present in an example of naturally occurring behaviour. Specifically, we focus on learned song in six populations of swamp sparrows in the Eastern United States. Using computational models, we assess the implications of conformist bias on the stability of these cultural traits.

Song in oscine songbirds is a well-studied example of animal social learning and cultural evolution[20,21]. To produce species-typical songs, young birds must memorise songs heard early in life, and later develop imitations of those songs. Evidence from hand-raised birds of several species, including swamp sparrows, demonstrates that individuals are capable of imitating model songs with great precision[22–24]. Laboratory studies also have revealed details of the process of cultural transmission[25]. Male swamp sparrows first memorise a set of syllable types (Fig. 1) during a sensitive period lasting the first 8 weeks of life[24]. The following spring, they themselves begin to sing and develop precise renditions of around 3 (range 1–6) syllable types. A key feature of this process is that individuals initially produce ~13 syllable types (around half of which are similar enough to be regarded as imitations or modifications of demonstrator syllable types), and this is gradually reduced to their adult repertoire (85% of which, in a sample of hand-raised birds, were similar to tape-recorded demonstrator syllable types)[22,23]. This process of overproduction and selective attrition has also been found in other species as well[25–28], and provides an extended period during which individuals can select which songs to produce based on those they hear others singing. This process makes song learning a particularly interesting case in which to examine conformist bias, which requires individuals to sample and compare more traits than they eventually produce[29].

Complementing laboratory experiments, field studies have demonstrated in some species of songbird that social learning might generate long-lasting traditions[30–33]. Many individuals within local populations may share the same song type[34], which suggests that learning is often precise[35,36]. Direct evidence for

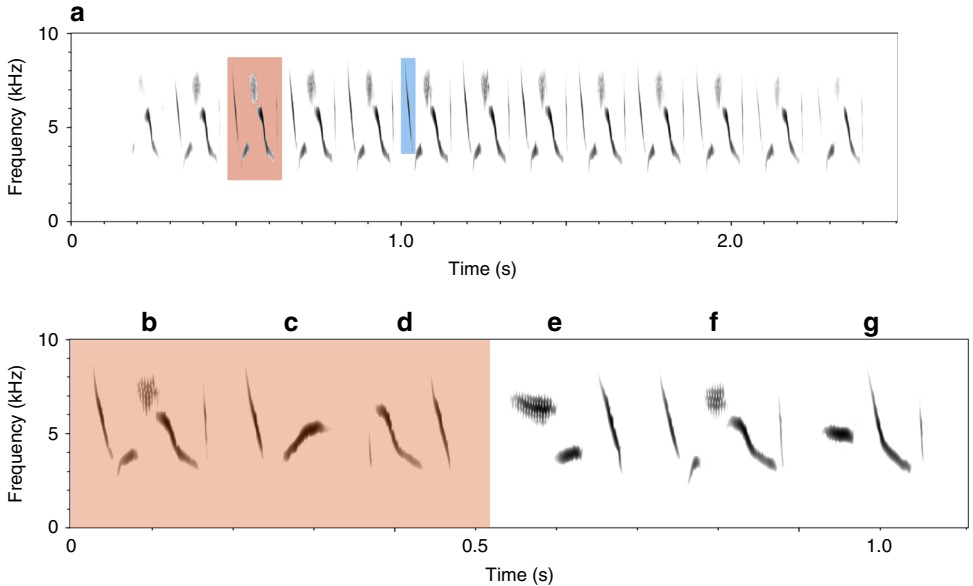

**Fig. 1** Swamp sparrow song structure. **a** Spectrogram of a swamp sparrow song sung by an individual from the Hudson Valley population. Swamp sparrow songs consist of one syllable (red) repeated 10 or more times. Each syllable consists of 2-5 elements or notes. **b–g** Examples of different syllable types. **b–d** Show the syllables that make up the repertoire of the same individual whose song is shown in **a**. **e–g** Show three syllable types sung by other males in the population, illustrating both the considerable diversity in syllable structure found within a population, but also how, as a consequence of vocal learning, different individuals also sometimes share the same syllable-type (**b**, **f**)

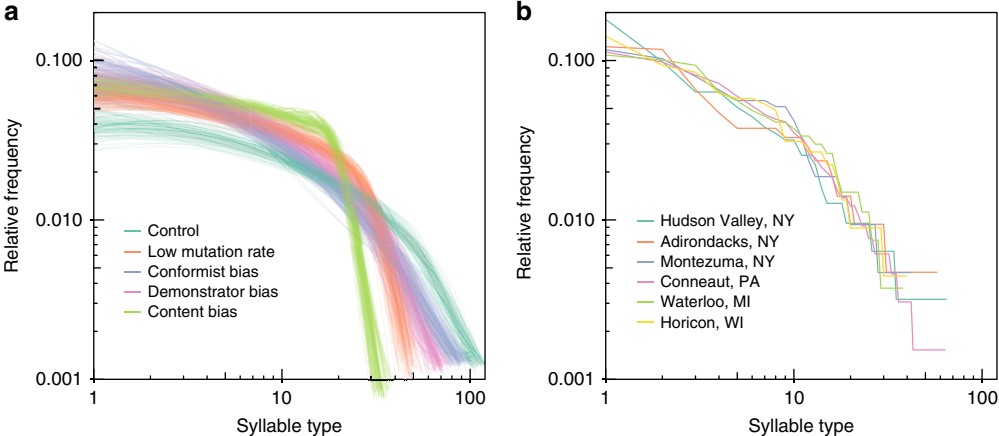

**Fig. 2** Simulated and empirical syllable type frequencies. **a** Frequency distributions of syllable types found in simulations with different types of bias, illustrating how different aspects of individual learning influence population-levels of diversity. A total of 100 simulations were carried out for each of 6 conditions. In the Control case, individuals had a mutation rate of 0.01 and no bias to learning. In the Low-mutation rate condition, $\mu = 0.001$. In the Demonstrator Bias simulations, $v = 2$. In the Content Bias simulations, $p_{att} = 0.1$. In the Conformist Bias simulations, $\alpha = 1.2$. **b** Frequency distributions of syllable types found in the empirical data set, from the 6 populations sampled (Conneaut, $n = 208$, Montezuma, $n = 71$, Adirondacks, $n = 70$, Waterloo, $n = 74$, Horicon, $n = 91$, Hudson Valley, $n = 101$), illustrating the consistency found between populations and suggesting a match between the empirical data and the simulations with Conformist Bias

stable traditions comes from diachronic studies showing that song types might persist for decades[30–32], but such studies are limited by the relatively recent advent of recording equipment. A recent analysis of songs in an isolated introduced population suggests that yellowhammer (*Emberiza citronella*) song types might persist for 100 years or more[33]. Neither laboratory nor field studies have provided detailed quantitative estimates of learning precision, learning strategies, or the longevity of song traditions, however, due to the challenging of inherently small sample sizes associated with such studies.

Here we use Approximate Bayesian Computation (ABC)[37,38] to overcome the latter problem, by fitting individual-based simulation models of cultural evolution to frequency distributions of syllable types from field recordings of 6 populations of swamp sparrows. ABC allows the estimation of posterior distributions of parameters for models that are too complex for likelihood calculations, but from which simulated data can readily be generated. It is thus particularly suited for studies of cultural transmission[39–41]. By pairing ABC with individual-based simulations[38], we are able to fit complex models in which individuals use multiple learning strategies.

We assessed the strength of evidence for three fundamental categories of social learning biases: conformist biases (described above), demonstrator biases, and content biases[8,42]. Demonstrator biases (sometimes referred to as "model" biases) occur when individuals prefer to learn from particular demonstrators (often referred to as "tutors" in the song-learning literature), irrespective of the trait they display. In the context of song learning, demonstrator biases might arise because some males sing more frequently than others, but also, hypothetically, if individuals prefer to learn from more successful males holding better territories. Content biases might occur when some variants of traits are inherently more attractive to learn than others regardless of who demonstrates them. In swamp sparrow song, previous studies have found no evidence that particular syllable types are inherently more attractive than others (see Supplementary Note 1), although some rare syllable types that we sampled appeared to deviate more from species-specific norms, which might suggest they are less likely to be copied, leading to content bias.

Each of these biases are expected to leave distinct imprints on observable frequency distributions of bird song syllables in populations[29] (Fig. 2a). Demonstrator biases reduce the effective population size, and thus reduce cultural diversity in a neutral model of evolution (much as bottlenecks reduce genetic diversity within populations), and hence increase the relative frequency of the commonest variants, and decrease the frequency of rare variants. Content biases result in many novel types in the population being ranked as unattractive, and being quickly removed from the population, and hence reduce the frequency of rare types in the population. Conformist biases increase the frequency of common variants. But if an individual can only sample a small proportion of the population, it is likely to sample both intermediate and rare variants only once, and thus rare variants are not disadvantaged much more than intermediate ones. All three biases might exist simultaneously, and thus our simulations took all three into account. But based on these observations (also see Fig. 2a), we expected slight differences in the consequences of the different biases, and thus to be able to differentiate between them and determine the strength of evidence for each.

## Results

**Syllable diversity and geographic variation**. We recorded the complete song repertoires of 615 adult male swamp sparrows from 6 different populations across the subspecies range of *M. g. georgiana* (see Supplementary Table 1 for details of the populations and samples). In these populations, swamp sparrows lived at high densities (typically > 150 km$^{-2}$). We then selected exemplars of each syllable type in each male's repertoire, based on a subjective assessment of recording quality, compared each of these syllables with every other syllable in the data set using the implementation of dynamic time warping in Luscinia (https://github.com/rflachlan/Luscinia), hierarchically clustered syllables, and used the Global Silhouette Index[43] to determine the number of types. This analysis found a total of 160 different syllable types in our sample (Supplementary Fig. 2). Within populations, there was no evidence of syllables being preferentially shared with neighbours (Mantel test of correlation geographic distances between individuals and Jaccard index of repertoire sharing, using populations as strata, $r = -0.0044$, $p = 0.28$), but syllable sharing

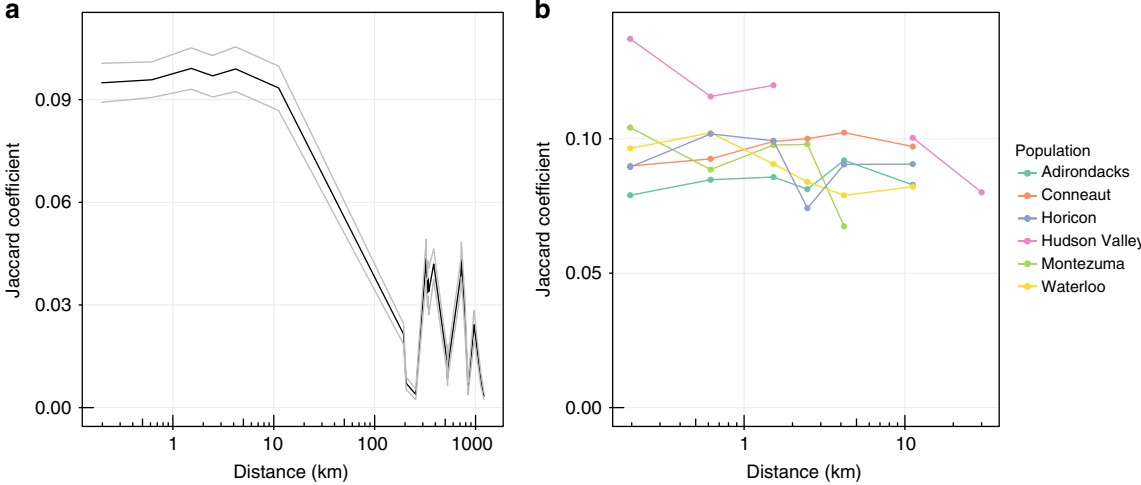

**Fig. 3** Geographic variation in syllable sharing. **a**, **b** Show the amount of syllable sharing between individuals' ($n = 615$) repertoires, measured using the Jaccard Index (proportion of shared syllable types out of all types found in a pair of individuals), against the geographical distance between them. **a** Syllable sharing was high within populations, with no significant within-population geographic structure, but much lower between populations. **b** Syllable sharing was similar between the six populations sampled

was much higher within populations than between populations (Fig. 3), and syllable sharing over the whole data set was clearly correlated with the distance between individuals (Mantel test of entire data set, $r = -0.307$, $p < 0.0002$). These results suggest that there is little or no geographic structure to demonstrator choices within a population, but that cultural divergence arises between populations. The frequency distribution of syllable types across the six population showed a long-tail distribution, with some very common types and many very rare types in each population. The distributions were markedly similar between populations (Fig. 2b).

**Cultural evolutionary models of song learning**. We next constructed an individual-based simulation model of syllable learning. Each run of the model lasts for a period of 5000 "years", and at the end of this period we measure the diversity of syllable types present in each population. In the model, there are a finite number, $N_s$, of possible syllable types, which are learned by individuals during the first year of life from adults that they selected from a population of $N_I$ individuals. Mutations occur at a fixed rate, $\mu$, per syllable-learning event, and generate a syllable type other than that produced by the demonstrator. "Mutations" encompassed innovations and errors, but also immigration of syllable types into the population. The mutated syllable type could already exist in the population, allowing for the possibility that individuals could re-invent syllables. Demonstrator biases in learning were incorporated by allocating each individual a value determining how attractive it would be as a demonstrator: the greater the variance, $v$, of this value, the greater the role of demonstrator bias. Content biases were incorporated by setting only a proportion of potential syllable types, $p_{att}$, as being attractive to learn. The smaller the value of $p_{att}$, the greater the proportion of novel syllables that were rated as unattractive, and thus the stronger the content bias. Finally, conformist biases were included by having individuals sample the repertoires of $N_T$ potential demonstrators, and memorise their syllable types. Then syllables were chosen relative to their frequency within the sample, raised to the power $\alpha$. Values of $\alpha$ greater than 1 correspond to conformist biases; values less than 1 to anti-conformist biases. These five key parameters ($\mu$, $N_T$, $v$, $p_{att}$, $\alpha$) each influence

the frequency distribution of syllable types (Fig. 2a). The aim of our analysis was to infer values for these parameters that were compatible with the empirical data.

We fit our model to the empirical data set using a Population Monte Carlo (PMC) variant of ABC[37], using a set of 13 summary statistics encompassing different types of information about the frequency distribution and pattern of sharing of syllable types, which we reduced to 6 components using partial least squares[44,45]. This way, we estimated that swamp sparrows in our recorded populations learn syllables with a mutation rate, $\mu$, of 0.0185 (95% Credible Interval: 0.0062–0.0465, see also Supplementary Fig. 3, Supplementary Table 3 for posterior distributions and parameter correlations). We did not find a strong correlation between population size and $\mu$ ($r = -0.202$), perhaps due to the role of conformist bias (Supplementary Note 2). Nevertheless, our credible intervals for mutation rate are likely influenced to some degree by our prior distributions of population size (log-uniform with limits 400 and 3000). This prior was informed by estimates of population size based on our surveys of the populations (Supplementary Table 1). It is harder to rule out extensive migration into and out of the population, and some fraction of the "mutations" in each population may be due to novel songs being introduced in by migrants. Thus, our estimate of $\mu$ is likely to be higher than the true value.

**Estimates of learning parameters**. We found no evidence to either rule in or out a strong demonstrator bias to learning, with a broad posterior distribution encompassing most of the prior distribution (median estimate of $v = 1.15$, CI: 0.0146–5.70). Our model does exclude the possibility of there being a very strong content bias ($p_{att} = 0.607$, CI: 0.217–0.973). Content bias was negatively correlated with mutation rate ($r = -0.612$): stronger content biases led to many innovated syllable types being removed from the population without being copied by others, and thus allowed a higher mutation rate to be consistent with the data. The lack of clear evidence for demonstrator and content biases does not mean that they do not exist. In the case of content biases, for example, it is well established that swamp sparrows selectively avoid learning heterospecific song[46]. But it appears that novel songs which fall outside the species-typical range are

only generated rarely, at levels at which the bias has only a minute effect on syllable frequency distributions.

We found strong evidence for a conformist bias, with the posterior distribution for parameter $\alpha$ clearly greater than 1 (median: 1.316, CI: 1.079–1.700). $\alpha$ was strongly negatively correlated ($r = -0.700$) with the number of potential demonstrators sampled, $N_T$ (median estimate of $N_T$: 4.71, CI: 2.59–23.3). The more demonstrators that individuals sampled during learning, the lower their error in assessing the frequency of syllable types and the weaker the conformist bias necessary to fit the data well. Interestingly, although our evidence for a conformist bias was clear, it was only of moderate strength at an individual level. For example, in the hypothetical situation of a male that sampled 5 demonstrators, each with three syllable types, with $\alpha = 1.316$, if 5 of the syllables were of one type and 10 of them unique, the probability of the male selecting the shared syllable type as the first syllable type in its repertoire would be 0.45 compared to a baseline probability of 0.33 without any conformist bias. At the population level, however, this bias had a large effect on the frequency distribution of syllable types (Fig. 2b) due to the amplifying effects of repeated cultural transmission. In a different context, a theoretical model predicted moderate or weak levels of conformist bias to be favoured when learning precision is high[47] (as it is in swamp sparrows), and moderate conformist biases were found in great tits that relied more on social learning for a foraging task[19].

**Estimates of the stability of syllable-type traditions.** Cultural transmission parameters influence the turnover of cultural traits, and thus the stability of traditions over time. Within our simulations, we recorded the date at which each syllable type was innovated, and then measured the age of all syllable types that were present at the end of the simulation run (types that persisted throughout the entire simulation run were assigned an age of 5000 years). The average age of the oldest syllable type in each population was 1537 years, and 8.6% of syllable types (and 26.5% of all syllables) were older than 500 years. More

common syllable types tended to be older than younger ones (Fig. 4). Thus, our results suggest that the individual song-learning behaviour of swamp sparrows is capable of maintaining traditions for extremely long periods of time. Our models cannot take into account large-scale demographic and ecological changes (e.g., the temporary extinction and re-colonisation of a marsh population due to changes in water level), instead these estimates merely suggest that, under stable conditions, syllable type traditions can last for a very long period. Direct anecdotal evidence for the maintenance of swamp sparrow syllable types comes from a visual comparison of syllable types in the Hudson Valley population with spectrograms from an earlier study with recordings made in 1976–1978[48] (31–33 years before our sample). All but two of the 19 syllable types sung by 2 or more males in our 2009 sample were found in the earlier sample, and the commonest type in 2009 was also the commonest type in the 1970's.

**An empirically-inspired model of conformist bias.** Our evidence for a conformist bias provokes a further question: what is the process underlying that bias? Social learning can be the result of cognitively complex processes (e.g., imitation), but also of simple ones (e.g., local or stimulus enhancement). We argue that the same is true for conformist biases, and that swamp sparrow conformity may not be based on sophisticated record-keeping of the relative frequencies of different syllable types, but rather on a much simpler process. The process of song overproduction and selective attrition described for swamp sparrow song development[23] provides such a process. In closely related species, it has been demonstrated that during selective attrition, individuals preferentially retain syllable types that they hear others singing[26,28]. This process of repeatedly comparing memorised syllable types against the vocal output of other individuals generates by itself a conformist bias, because common syllables are more likely to be encountered repeatedly.

We modelled this process as follows: individuals first memorise a sample of $N_{T1}$ adult syllable repertoires. They select their adult repertoire from this memorised sample by removing syllable types, one by one. At each stage, the decision of which type to remove is determined partly by a second random sample of $N_{T2}$ adult repertoires: the more frequently syllable types occur in this second set, the less likely they are to be removed. This process is in line with the idea that birds listen to adults singing around them during this period of selective attrition, and are less likely to remove a syllable type that they have heard recently (note that the lack of geographic structure within populations suggests that this process may often occur before learners acquire their own territory). This process of repeated sampling inherently favours common syllable types over rare ones. We quantified the relative role of the first and second sampling periods during selection with a weight parameter, $W$. Repeating the fitting process with ABC, we found that this model too was readily able to closely fit the empirical data, with parameter estimates of $N_{T1}$: 3.08 (CI: 2.53–4.42); $N_{T2}$: 3.6 (CI: 1.6–10.2); $W$: 1.58 (CI: 0.54–9.60). $N_{T2}$ was strongly positively correlated with $W$: the more syllables were sampled during the second phase, the greater the relative role of the first phase. This relationship effectively maintained a constant ratio between the contributions of the first and second phase of learning to the overall choice of which syllable type to retain in their repertoire. The frequency distribution of syllable types generated by the model under these parameters showed the same relatively high levels of common and rare syllable types as found in both the conformist bias model and the empirical data.

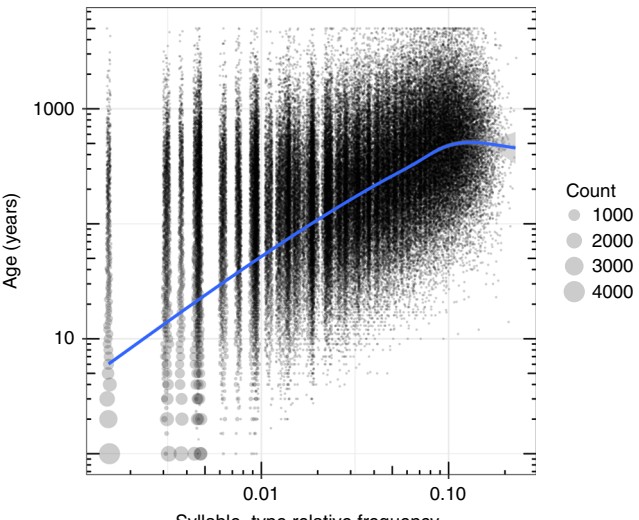

**Fig. 4** Age of syllable types in fitted simulations. The age of syllable types in the 1000 simulations that best fit the empirical data. Each point represents a combination of age in years since the syllable type was innovated and frequency within the population (point size represents the number of times that combination arose). More common syllable types tended to be older, and many syllable types are estimated to be older than 500 years

## Discussion

We have for the first time been able to derive estimates of the individual behavioural parameters underlying cultural transmission of bird song. Our ABC-fitted models suggest that song learning in swamp sparrows is highly precise. This precise learning is complemented by a conformist bias, which may be generated by the cognitively simple process of overproduction and selective attrition. This hypothesis could be tested with further laboratory experiments investigating how songs heard during the motor phase of song development influence the process of selective attrition and influence the choice of which type is learned. Together, these conditions of precise learning and conformist bias would allow song traditions to persist for many hundreds of years, and if our model is an accurate reflection of swamp sparrow learning, it would demonstrate that non-human cultural traditions can match the stability of those found in humans.

Our methods can readily be applied to other species of vocal learners for which similar data sets exist. Among songbirds, it is clear that some species of birds learn precisely, such as swamp sparrows, while others rarely learn all parts of a demonstrator's song precisely[49]. Yet, because we have lacked methods to quantify the precision of learning, we have not been able to apply comparative methods to examine the evolutionary causes and consequences of song-learning behaviour. Doing so would allow us, for example, to test hypotheses about the role of sexual selection in the evolution of precise learning[50] or how learning might influence speciation processes[51].

Previous laboratory experiments have demonstrated that swamp sparrows are capable of imitating the songs of tutors with high precision[22,23,46]. Our results suggest that nearly all songs are learned with such precision, which in turn indicates that swamp sparrows learn syllables as precisely as possible, and avoid singing unique or rare syllable types. This is consistent with experimental evidence that more typical variants of common syllable types are more attractive to female swamp sparrows, and stimulate a more aggressive response from territorial males[52]. The consequences of song-learning precision serving as an assessment signal[52–54] may have been selection for precise transmission but also for a simple mechanism of conformist transmission, and, in turn, stable traditions. In human culture, precise transmission and stable traditions have been argued to be a prerequisite for complex, cumulative culture[1,3,4]. In swamp sparrow song, studies of sub-syllabic note type variation have revealed overlapping and context-dependent patterns of categorical perception that suggest an unusual degree of sophistication and hierarchical structure in cultural evolution[55–57]. But despite possessing precise transmission that leads to stable traditions, the complexity of swamp sparrow vocal culture of course does not begin to approach that of humans. Our findings, then, suggest that the ability to transmit traditions with precision can no longer be considered a fundamental difference between how human and non-human cultures evolve.

## Methods

**Song recordings.** We recorded the song repertoires of 615 adult male swamp sparrows between 05 May 2008 and 12 July 2009 from six populations across north-eastern USA (Supplementary Table 1). All populations are within the species range of the subspecies *M. g. georgiana*. We recorded the location of each individual's territory using a hand-held GPS device. Recordings were made with a Sony PCM D50 digital recorder at a sampling rate of 44.1kHZ and at 16bits, using a Shure SM57 microphone mounted in a Sony PBR 330 parabola. To maximize the chances of recording each of the syllable types in a male's repertoire, we recorded each male for either at least one hour or until it cycled through its supposed repertoire at least 1.5 times[57,58]. We selected one exemplar for each syllable type in each individual's repertoire, based on a subjective assessment of recording quality.

**Song measurements.** Exemplars were measured using the Luscinia sound analysis program (http://github.com/rflachlan/luscinia, version 2.17.11.22.01). Spectrograms were produced after high-pass filtering the recording at 1 kHz. We standardised spectrogram settings for our recordings. The window length was 5 ms, and the time step was 0.5 ms. A Hamming window function was used. Recordings were further processed using the dereverberation function of Luscinia with a 'dereverberation range' of 50 ms and 'dereverberation' of between 25 and 100%, checking visually that the signal was not attenuated by the algorithm. We set the dynamic range to −40 dB, relative to the maximum amplitude within the entire spectrogram and set the dynamic equalisation function to 200 ms to increase the amplitude of the quieter the initial and final syllables within swamp sparrow songs. Using Luscinia, we then measured each element, and marked each syllable within each exemplar.

**Syllable comparison.** We compared the structure of each syllable with every other syllable using the dynamic time warping (DTW) algorithm in Luscinia (see[52,59] for more details), setting the Compression Factor parameter to 0.001, with a Minimum Element length of 10. This ensured that every element in the data set was characterised by 10 data-points, evenly spread through its length. We used time, fundamental frequency, fundamental frequency change, and vibrato amplitude as the basis for the DTW comparisons (with relative weightings 10, 1, 1, and 0.25). Fundamental frequency was log-transformed.

The outcome of the Luscinia comparison is a matrix of syllable dissimilarities. We clustered syllables on the basis of this matrix using the hierarchical UPGMA algorithm, and assigned syllables to types on the basis of the resulting dendrogram. We calculated a measure of clustering tendency, the global silhouette index (GSI)[43] at each depth within the dendrogram, and searched for peaks in the index, which identify natural clusters among the syllables. This clustering procedure produced a dendrogram with a peak in the GSI at $k = 160$ syllable type clusters (Supplementary Fig. 2). These clusters closely matched labels given to syllables by subjective visual assessment of spectrograms for the Conneaut population (73 human vs. 64 DTW clusters; Normalised Mutual Information index of cluster similarity[60] = 0.868). We used this clustering solution for all further analyses.

**Analysis of geographic variation.** In order to verify our assumption that any particular adult within a population would be equally likely to be the demonstrator for any particular juvenile, we first tested whether, within populations, syllables tended to be shared with territorial neighbours or not. Using the $k = 160$ clustering mentioned above, we measured the Jaccard coefficient of repertoire sharing between each pair of individuals in the population, and binned comparisons based on the geographic distance between them. We also carried out Mantel tests of correlation between matrices of Jaccard coefficients and geographical distance.

**Cultural evolutionary models of song learning and conformity.** We examined two different models of song learning, both of which included a conformist bias: in model 1, this bias was defined explicitly; in model 2 it arose from the process of selective attrition. We define a conformist bias as any tendency to copy common traits at an even greater probability than their frequency in the population. In most cases, however, it is impractical for individuals to assess trait frequencies in an entire population. A more realistic proposition is that individuals sample a subset of the population, and select more frequent traits from that sample. Simulations were initiated with syllable types randomly allocated to individuals, and lasted 5000 years, at the end of which summary statistics were calculated.

**Model 1.** We constructed an individual-based simulation model of song learning and cultural evolution. Each population consisted of $N_p$ individuals. Each individual in the model learned song syllables during its first year of life only. Syllables were characterised by an integer, $x$, (range: $1$-$N_s$). Each syllable type had an intrinsic attractiveness to be learned, $M_x$, which was determined at the outset of the simulation and remained constant throughout the simulation (with a proportion of syllable types, $p_{att}$, deemed attractive with $M = 1$, and the remainder unattractive with $M = 0.05$). Each individual, $m$, was first assigned a repertoire size $r_m$ by sampling randomly from the distribution of repertoire sizes in our empirical data set. Then, juvenile individuals filled their repertoire by imitating adults in the population. In each year of the population, individuals had a mortality rate of 0.4 (our results were not sensitive to this parameter). We used years as convenient time-steps for the model since swamp sparrows have one clear breeding season per year, and learn songs only during a fixed period during their first year of life. Individuals that died were immediately replaced. Each male in the population was also assigned at birth a "demonstrator attractiveness index", $t_m$. This was calculated as: $t_m = e^{\varphi(0, v)}$, where $v$ is a parameter that determined the variability in demonstrator attractiveness in the population.

The first step of imitation was to sample a set of $N_T$ potential adult demonstrators from the population and memorise each syllable in their repertoires. If a given syllable type had already been memorised, then it was not added to the memory. But the frequencies of each syllable type in the sample of demonstrators, $F_x$, was recorded. By raising $F_x$ to a parameter α, conformist (α > 1) or anti-conformist (α < 1) biases could be created, where the higher α was, the stronger the bias. Some syllable types were produced by multiple demonstrators. Therefore, in

order to model the effects of variation in demonstrator attractiveness, we had to produce an aggregate demonstrator score for a particular syllable, across all the demonstrators sampled by a particular learner. The simulation thus calculated a combined demonstrator attractiveness score, $T_x$, for each syllable type, $x$, in the male's memory, by taking the mean of the demonstrator attractiveness scores, $t_m$, for each demonstrator that sang that particular type. In the main model, the probability of a male selecting a given syllable type, $x$, to learn was then set proportional to: $Pl_x = F_x^\alpha \cdot M_x \cdot T_x$

Syllables were learned by sampling from the distribution of Pl without replacement. With a probability μ, individuals produced a different syllable type from that selected ("mutation"). In this case, the syllable type was sampled randomly from the uniform distribution {1:$N_s$}.

**Model 2**. In the second model, learning occurred over two phases. In the first phase, the repertoires of $N_{T1}$ adult demonstrators were memorised, just as in Model 1. In the second phase, the repertoires of a separate sample of $N_{T2}$ adults were memorised. Syllable types that were encountered for the first time in this second phase could not be memorised (i.e., birds had to select from the list of syllable types they memorised in the first phase). The frequency of syllable types encountered in the second phase that had already been memorised in the first phase was given by $G$. Then, for each syllable type $x$: $Pr_x = (W + G_x) \cdot M_x \cdot T_x$

Individuals then progressively reduced their repertoire size from that memorised in the first phase: in each step of this process, syllables were selected by sampling without replacement from Pr until one syllable remained unselected, and that syllable was then removed from the repertoire. If more than 8 syllable types had been memorised in the first phase, the very first reduction step reduced the repertoire to 8. This approach was required to make the simulation computationally feasible, but it also corresponds to the maximum number of imitated syllable types observed in songs produced by swamp sparrows during the early "plastic song" phase of song development[23].

**Summary statistics**. To compare empirical to simulated data, we measured 13 summary statistics from each population. Inference in ABC is subject to the summary statistics used, and so we used an exhaustive approach, employing as many statistics as possible. These were as follows: (1) Singletons: proportion of syllable types that were sung by only one individual in the sample; (2) Rare syllable types: proportion of syllable types that were sung by 4 individuals or less; (3) Intermediate-frequency syllable types: proportion of syllable types that were sung by more than one but less than or equal to $c$ individuals, where the threshold $c$ was 5% of the total number of syllable types in the sample; (4) Common-frequency syllable types: proportion of syllable types that were sung by more than $c$ individuals; (5) ns, The number of syllable types in the sample; (6) The number of individuals singing the most common syllable type in the sample; (7) $H$, an index of diversity, calculated as:

$$H = \sum_i^{ns} f_i . \log_2(f_i),$$

where $f_i$ is the proportion of syllables in the sample of type $i$. (8) $\alpha_p$, the exponent of the fitted power-law function. We fitted a power-law function to the ordered empirical probability distribution of syllable types. $\alpha_p$ was calculated[61] as follows:

$$\alpha_p = 1 + ns . \left( \sum_i^{ns} \ln(2.f_i) \right)^{-1}.$$

(9) The degree of fit of the cumulative empirical probability distribution to the fitted power-law distribution; specifically the Kolmogorov–Smirnov statistic (the largest absolute discrepancy between the empirical and power-law distribution). (10) A second statistic capturing the fit of the data to a power-law distribution: the absolute discrepancy between the two when the frequency of the syllable type is 2. This captured the fit of the power-law distribution at one extreme of the distribution, where we had observed key departures in exploratory work. (11) The proportion of pairs of individuals that shared at least one syllable type. (12) The proportion of pairs of individuals that shared more than one syllable type. (13) The mean correlation-coefficient between syllable types:

$$\bar{r} = \frac{\sum_i^{ns} \sum_j^i r_{i,j} . \sqrt{p_i . p_j}}{\sum_i^{ns} \sum_j^i \sqrt{p_i . p_j}}, \text{ where } r_{i,j} = \frac{p_{i,j} - p_i . p_j}{\sqrt{p_i \left(1 - p_j\right) p_j \left(1 - p_i\right)}},$$

where $p_{i,j}$ is the probability of types $i$ and $j$ occurring in one bird's repertoire, and $p_i$ is the probability of type $i$ in the sample. Statistics (12) and (13) both capture information relating to the probability that individuals share more than just one syllable type in their repertoires, and are thus intended to be related to a tendency to learn songs from one particular demonstrator. Statistics (1–4, 11, 12) were logit-transformed.

To address potential redundancy among the 13 statistics, and to find the most informative combination of statistics, we used a partial least squares method (PLS)[44]. First we simulated data, drawing 2400 sets of parameter values from priors (see below) and running a simulation for each set. For each simulation run, we measured the 13 summary statistics described above. We then ran a partial least squares analysis, using the simulation parameters as dependent variables and the summary statistics as independent variables, using the package pls[45]. A leave-one-out cross validation suggested that 6 components were best able to explain variation in the simulation parameters on the basis of the summary statistics (Supplementary Fig. 4, 5). The loading weights for these components are given in Supplementary Table 2.

**Approximate Bayesian Computation**. Approximate Bayesian Computation (ABC) attempts to provide an unbiased estimate of the posterior distribution of parameters ($p(\theta|y)$ by comparing summary statistics taken from empirical data ($y$) with summary statistics taken from simulations of the empirical data ($y_s$): ($\theta|y$)∝ $Pr(|y_s - y| < \varepsilon|\theta) \cdot p(\theta)$. Simulations are carried out with parameters drawn from prior distributions ($p(\theta)$), and if the summary statistics calculated from simulations are within a threshold ε of the empirical summary statistics, the parameter values are accepted. A distribution of such accepted parameter values constitute an unbiased estimate of the posterior distribution in the above equation, as $\varepsilon \to 0$. (see ref. [37] for a general review). In each run of our simulations, we carried out independent simulations for each of the 6 populations. The overall dissimilarity between the simulated and empirical PLS dimensions was calculated as the Euclidean distance, over all PLS dimensions, between the mean of the simulated PLS scores and the mean of the empirical PLS scores.

Since the basic ABC approach can require a very large number of simulations, and since our agent-based simulations are relatively computationally taxing, we employed the Population Monte-Carlo variant of ABC (PMC-ABC,[37]). PMC-ABC increases the efficiency of ABC by carrying out a series of sets of simulations in which the threshold ε is progressively reduced. We used the series {20, 16, 13, 11, 9, 8, 7, 6, 5, 4, 3, 2}. In each set of simulations, we obtained 1000 accepted parameter sets. Because model comparison in ABC is problematic[62], we adopted a strategy that allowed us to test alternative hypotheses within models.

**Prior distributions**. We set the prior distributions for $N_p$, $N_T$, $N_s$, $v$, $\alpha$, and $\mu$ as log-uniform distribution with limits $N_p$:{400, 3000}; $N_s$:{180, 500}; $N_T$:{2.5, 50}; $v$:{0.01, 6}; $\alpha$:{0.25, 4}; $\mu$:{0.0001, 0.3}. $p_{att}$, was given a uniform distribution {0.01,1}. In model 2, $N_{T1}$, $N_{T2}$, and $W$ were also given log-uniform distributions with limits $N_{T1}$:{2.5, 15}; $N_{T2}$: {1.5, 50}; $W$: {0.01, 20}. These choices were made to encompass all potential outcomes (e.g., from very strong demonstrator biases to very weak ones) and to be uninformative.

**Validation of ABC**. We carried out a leave-one-out cross validation analysis to check that our ABC design was providing unbiased parameter estimates. We simulated 100,000 samples directly from the priors, and, taking each sample in turn, estimated parameter values using the remaining samples. Because number of demonstrators sampled, $N_T$, was strongly correlated with conformist bias, we fixed it at 5. This analysis showed that the two key parameters, conformist bias and mutation rate, were accurately estimated by the analysis (Supplementary Fig. 6). Demonstrator bias was not accurately estimated by the analysis, while content bias was only estimated precisely when it was strong ($p_{att} < \sim 0.25$). We note that these two latter parameters were included in our model only because of their potential confounding effects on the parameters of primary interest.

**Code availability**. The simulation code is freely available at https://github.com/rflachlan/SongABCJPPF.

**Data availability**. The Luscinia database of measured swamp sparrow songs used in the recording can be found at: https://doi.org/10.6084/m9.figshare.5625310 and includes all recordings used in the analysis; metadata (including locations of individuals); and song measurements. The fileset also includes the output of the song comparison analysis, and the output of the ABC analysis.

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

## Acknowledgements

Funding was provided by the Duke University Provost Office and U.S. National Science Foundation grant IOS-1144991 to S.N.; and the Bill & Melinda Gates Foundation OPP1084362 to O.R. Permission and assistance with fieldwork was kindly provided by Montezuma National Wildlife Refuge, Horicon NWR, Thompson Ponds, Phyllis Haehnle Memorial Sanctuary, NY Department of Environmental Conservation, Hudson River National Estuarine Research Reserve, the Pennsylvania State Game Commission and the Pymatuning Laboratory of Ecology. We thank J. Cooper and M. McMahon for providing helpful comments on our manuscript.

## Author contributions

R.F.L. conceived the study, recorded and measured the songs, designed and implemented the simulations and analyses, and wrote the paper. O.R. contributed to the design of the ABC analysis and the writing of the paper. S.N. contributed to the design of the study and to the writing of the paper.

## Additional information

**Competing interests:** The authors declare no competing interests.

