## [Peer Review File · Nature Communications]

Reviewers' comments:

Reviewer #1 (Remarks to the Author):

This paper takes an admirably multifaceted approach to the study of song evolution in swamp sparrows and demonstrates that conformist biased cultural transmission can occur in songbirds. Lachlan et al. combine empirical data with computational modeling to demonstrate that syllable types are likely to be preferentially produced by swamp sparrows when they are already common in the population (conformist bias) and that syllable types can persist for hundreds of years.

I have a few concerns for the authors to consider. First, I think that the claim that this is the first demonstration of conformist bias in natural animal behavior is perhaps overstated and can be given a little more nuance. There are more references to be cited around lines 42-45, particularly in the primate literature, e.g.:

Whiten, A., Horner, V., & De Waal, F. B. (2005). Conformity to cultural norms of tool use in chimpanzees. *Nature*, 437(7059), 737-740.

Haun, D. B., Rekers, Y., & Tomasello, M. (2012). Majority-biased transmission in chimpanzees and human children, but not orangutans. *Current Biology*, 22(8), 727-731.

van de Waal E, Borgeaud C, Whiten A. (2013). Potent social learning and conformity shape a wild primate's foraging decisions. *Science* 340, 483-485.

Whiten, A., & van de Waal, E. (2016). Identifying and dissecting conformity in animals in the wild: Further analysis of primate data. *Animal Behaviour*, 122, e1-e4.

Dindo, M., Whiten, A., & de Waal, F. B. (2009). In-group conformity sustains different foraging traditions in capuchin monkeys (*Cebus apella*). *PLoS One*, 4(11), e7858.

Also, a better citation than the review by Morgan and Laland might be [Pike, T. W., & Laland, K. N. (2010). Conformist learning in nine-spined sticklebacks' foraging decisions. *Biology Letters*, rsbl20091014.] since the Morgan and Laland review states "The only study of which we are aware that provides clear evidence of non-human animals exhibiting a disproportionate tendency to adopt the majority behavior is Pike and Laland's (2010) investigation of public information use in sticklebacks. Given the taxonomic distance between fish and humans, this finding is most likely to reflect convergent selection for conformity rather than a homologous capability (Laland et al., (2011b))."

Also, the cited Aplin et al. paper is indeed an experiment, but it is one in which a conformist tradition arose naturally in the cultural transmission of an experimental trait (initial association of a color with food availability in a bird feeder led to a transmitted color preference that persisted even when both colors were later associated with food).

Similarly, in the discussion, the last sentence ["Our findings, then, suggest that the ability to transmit traditions with precision can no longer be considered a fundamental difference between how human and non-human cultures evolve."] might be overstated. The authors don't really make a convincing case that researchers in the relevant fields believe that precise transmission is unique to humans.

Also, I liked that the authors proposed a mechanism for the conformist bias: if birds learn more syllables than they need in a frequency-dependent way and then are more likely to prune rarely heard syllables from their repertoire, an over-representation of frequent syllables might follow. Is there a

meaningful distinction to be made, though, between conformist bias in humans, in which conformity might be the actual goal, and this compounding of multiple frequency-dependent processes in birds? I'm asking genuinely; I'm not sure.

Other things:

I think the paper could better conceptually explain the difference between tutor attractiveness and song attractiveness. Similarly, I conceptually understand model bias, conformist bias, and content bias, but I wanted an explicit comparison of the three and how to distinguish them in practice, especially since they seem like they might be correlated with one another and also occur in combination.

Could territory size affect a juvenile's exposure to song? If he grows up on a large territory, is he less likely to hear other conspecifics?

I am not sure about the analysis of geographic variation. The methods section does not say what statistics the authors would calculate to accept or reject their hypothesis. It seems like one could do a Mantel test or a spatial autocorrelation analysis given the pairwise values. Based on Figure 2 and the text, the analysis seems qualitative and not quantitative, so I was unsure of the evidence in question when the authors said there was no evidence of preferential syllable sharing with neighbors. Also, the use of kilometers in panel A and meters in panel B made me not immediately realize that the left side of panel A was probably an average of panel B. How large is a typical swamp sparrow territory?

Terminology: Some of the song terminology is presented in a confusing way. For example, syllable type and song type appear to be used interchangeably (see Figure 1 caption vs. axis labels, but it also occurs in the text), which is perhaps a shorthand that can apply to swamp sparrows, but it does not scale up to many other species and so could be confusing to those who are familiar with other singing styles. In addition, the words "exemplar" and "element" should be defined in the context of this paper.

Could the spectrograms of one swamp sparrow's full repertoire be included and labeled with the relevant terms?

In this type of ABC model, do timesteps in the simulation always represent years? The authors could include a few sentences to justify the assumption that one timestep in the model represents a calendar year.

Figure 1. What would the authors like the reader to take away from panels B and C? Just a qualitative sense that the real data look middle-of-the-road and probably most like conformist bias?

Figure S1. The peak used to determine the chosen k is not very sharp, with 125 to 200 being pretty similar in GSI. Is this typical for this type of analysis?

Figure S2. Could you show the prior distribution as well?

Minor issues:

Line 34: More than two citations might be good to illustrate the depth of study on conformist transmission.

Line 55: approximately half of the syllable types are similar to tutor songs; where does the other half come from?

Line 90: The previous studies mentioned in this sentence should be cited.

Line 94: "the presence or otherwise of conformist biases" – does otherwise here just mean absence? Or does it imply different degrees?

Line 109-112: This is reminiscent of a site frequency spectrum in genetics, and much work has been done to quantify the effects of different forms of selection on these distributions. It might provide an interesting point of comparison.

Line 120 and elsewhere: I found it confusing to have the tutor's song called "model" in the context of the simulation model. For example, I initially read "model biases" to mean "biases in the [simulation] model," but that is not what the authors mean.

Line 129: should this say Fig 1b?

Line 159: "date at which each syllable type was innovated" – how do the authors account for the initial syllables present, or for the lifespan of syllables that are still present at the end of the simulation?

Line 196: I'm not sure what the authors mean by counterbalance here.

Line 203: Does "our ABC-fitted model" refer to Model 1 or Model 2 or both?

Line 217: Need citations for human culture sentence.

Line 267: are the results of this human visual assessment available?

Line 306: How is alpha chosen? Does it matter how much bigger (or smaller) it is than 1 for the results, and if so, was it estimated from the data or randomly assigned?

Line 383: "The overall dissimilarity between the simulated and empirical PLS dimensions was calculated as:" it seems like an equation is missing after the colon.

Reviewer #2 (Remarks to the Author):

This paper is novel, exciting, and the results appear to be robust. I'm also sure that the methodology will inspire future analyses of similar data sets of bird song. The data-set which underpins the paper also appears to be robust and very impressive (615 individuals in 6 populations). In general, I only have minor comments. In particular, I'm pleased that the authors have extended their findings to all social learning in animals (as opposed to only addressing the bird song literature), and I think it will be read with interest by researchers interested in the evolution of human and animal culture more broadly. However I would highlight that in doing so, the authors need to be more careful – in particular there are some sections of the introduction and discussion that need more careful consideration of existing literature and controversies. I discuss this further below.

Abstract – "Our results demonstrate conformist bias for the first time in natural animal behaviour..." This is over-reaching. For example, studies of stone-tool use in chimpanzees have claimed to find evidence for conformist bias in the way in which females change their techniques after immigrating into differently behaving groups. Also see evidence in sperm whales (discussed below). I would just shorten this statement to "Our results demonstrate conformist bias in song learning and show that this, ..."

Introduction – The first two paragraphs of the introduction are a confusing mix of the underlying conditions for cumulative culture and for stable traditions. The current literature argues that cumulative cultural evolution (CEE) requires high fidelity copying (e.g. imitation) and progressive innovations that incorporated into traditions (e.g. through pay-off biased learning). Stable traditions may require relatively 'error-free' precise copying (although not necessarily), and may be facilitated by conformity.

In these two paragraphs, the authors state that CCE is possibly not observed in other animals because traditions are not stable enough, and that humans may also be unique in using social learning strategies. The second of these two statements is demonstrably false – multiple other species have been shown to use social learning strategies (although pay-off biased and conformist strategies may possibly be more restricted). The first, as stated above, conflates the conditions thought necessary for CCE and stable traditions.

I suggest the authors reword these two paragraphs and focus on stable traditions rather than CCE), and make it clear that they are referring to imitation (high fidelity copying) and conformity (making traditions resistant to erosion from errors or from immigration).

L34 – I suggest changing “strategies for” to “biases in”, to make the causation a bit less explicit.

L43 – I would also reference van de Waal et al. (2013) Science here. This paper found some evidence for conformity in vervet monkeys.

L44-45 – Conformity in natural behaviour has arguably been shown in primates. Additionally, Cantor et al. (2015) Nature Communications, used a similar simulation based approach in sperm whales, where they modeled data with various biases to produce patterns that they compared with real recorded vocal dialects in whales. They also found that the observed patterns were likely to have arisen through conformist learning (along with homophily).

L55 – I’m interested in where the other half of the dialect comes from? Are these an innate syllable set, or innovated by each juvenile?

L61 – reference?

L78 – “it is thus particularly suited for studies of cultural transmission” ABC is used for a range of applications; can you justify this statement more explicitly? Also, be aware that there is some current controversy about whether population-level data can be used to look for conformity (Acerbi et al. scientific reports, and others, but see Smaldino et al. Bioxiv). I personally think the approach taken here is fine, but the authors should be clearly cognizant that this argument exists.

L85 – This is not exactly what people will think of as a classic ‘model bias’. I would also include an example more familiar to the readers. For example, ‘copy related individuals’.

L95 – This is a funny way to frame it the rest of the paper. I would instead just state that you are investigating whether they show evidence of biased transmission, and if so, identify what biases.

Results – The figures aren’t referred to in order, and I found this very confusing. Can you please reorder them in a more logical fashion? I also suggest a new figure in the main text that summarizes the data collection; showing the six populations and the geographic distance between the samples collected within and between populations. This could combined in a composite figure with figure 2, which surely should be presented first (and perhaps also with the syllable examples in figure 1a).

L140 – In light of other studies showing evidence for preferences for species or subspecies song, it is surprising there is no evidence for content biases. Worth going into more depth on this result, perhaps in the discussion?

L145-150 – I find the link between sampling extent and strength of conformity unsurprising, as these two would work together? That is, there is less error in a larger sample, so even a weaker bias will move the population faster towards uniformity. However I agree it is important to highlight this.

L163 – I would qualify this statement a little more, as you don’t show this directly. Even just stating “Thus the results of our analysis suggest that ... is potentially capable”.

Discussion – What is the proposed experiment? What should be tested next? I think it would add a lot to your discussion if you detailed what the next steps would be in validating or extending these

findings.

L208 – This is an overstatement of your findings. You don't have direct evidence of the past conditions of the populations, only the current distribution of song-types. I don't think your evidence is stronger than, for example, primate archeology that has found stone-tools dating back 1000s of years, or even the New Zealand studies on song of European songbirds.

L222 – "The complexity of song sparrow vocal culture of course does not being to approach that of humans". This qualification isn't necessary, of course it is the case. I don't think the work benefits much from these sorts of direct comparisons with humans. I would instead focus on animal culture, and this emerging field (including the exciting evolutionary and ecological implications of animal culture).

L224- again, I don't think it benefits from trying to compare human and non-human culture. This feels like an artificial insertion – imitation is well known in bird song, and you don't discuss imitation much previous to this. Why don't you instead focus on conformity, or on how these results suggest that populations can maintain deep learning-based divisions in behaviour, with evolutionary/ecological implications?

L238 – did you consider the relative frequency of song types within each male's complete repertoire?

Reviewer #3 (Remarks to the Author):

Review of Cultural conformity generates extremely stable traditions in bird song
Submitted to Nature Communications

This paper presents a big data/Bayesian model analysis of cultural evolution and social learning in a well-studied songbird, the swamp sparrow. Using a large set of recordings of contemporary swamp sparrow males and a model of social learning, the authors use an innovative approach called Approximate Bayesian Computation to estimate the degree of fidelity among and between populations, the expected length of time that long-lived syllable types would persist in a population, and to evaluate models of transmission bias. They conclude that song traditions in this species may be very long-lived: up to roughly 500 years. If correct, this implies that a very widespread claim (based mainly on studies in nonhuman primates) that only human social learning leads to long-lived cultural traditions, is incorrect.

I believe that this is an important message, supported by a strong analysis in a well-understood species. I also think the results would be of interest to a large group of scientists including social scientists, linguists, anthropologists and computer scientists as well as biologists, and thus think it has the breadth to warrant publication in this journal. Thus I strongly support publication. My suggestions below are intended to strengthen the conclusions and broaden the impact.

General points:

1. The biggest issue with the acceptance of any modelling study, regardless of how much data underlies the model, is always the issue of its relationship to reality; this study will be no exception. Taking the viewpoint of a sceptic (and there will be many among the non-biologist target groups listed above), one can always argue that the conclusions follow from the parameters chosen for the model, or that the data do not constrain the model enough.

I think the authors would be able to strengthen their message if they can offer some independent indication that swamp sparrow traditions are indeed long-lived. For example, are there old-recordings from fifty or more years ago, that are properly classified in this cluster hierarchy? (or old spectrograms my old copy of "Birds of North America" from 1966 has spectrograms strikingly similar to those in this paper). I'm not asking for a detailed analysis, but more of an "eyeball what data is available" approach in the Discussion.

It might also be mentioned that even for the sounds of language, we have only writing to push us back to 500 years, and we just assume that this reflects sound: but detailed speech analysis of vocal dialectal details change within a single lifetime (see Harrington et al 2000)

Alternatively or in addition, some measure of the fidelity of learning from lab studies with tape tutored animals would help strengthen this core argument. This could simply reference the large literature for this species, most of it by Peter Marler et al.

2. I think a few more words about the approach should be placed in the main part of the paper (not just the methods). Crucially, it is implied (e.g. line 156) but never stated that the models underlying these simulations are run over many generations, but neither this nor the number of generations simulated is specified. Nor is the mapping from generations to years (though I guess the assumption is that this is one to one would be valid for this seasonally breeding case).

Also, perhaps a sentence about the main differences between the ABC approach and the more standard Bayesian methods (like MCMC-based model testing) would be welcome, and a specification of the advantage(s) of this new approach.

3. I don't think the authors try hard enough to connect this work to the long-running "animal culture" controversy (perhaps by design, given the controversy!). A lot of this work focusses on non-vocal social learning (e.g. primates learning tool use from each other in the work of Whiten and colleagues) and shows poor fidelity and rapid decay; this has led many technology-focused theorists like Boyd and Richerson to the conclusion that high-fidelity, lasting traditions and a cultural ratchet effect are uniquely human.

But this literature often ignores the work on vocal traditions, both in birds (which are well-covered in the current reference list) and in humpback whales, where the work of Noad and Garland could be cited.

4. Minor typos/odd words:

line 204: some of the mutation rate -> some component of the mutation rate

209-210: it can no longer be considered... : the phrasing here is quite awkward and roundabout - please rephrase.

Line 534: aspects to individual learning: aspects of...

Refs:

Harrington, Jonathan, Sallyanne Palethorpe & Catherine Watson. 2000. Does the Queen speak the Queen's English? *Nature* 408, 927-8.

Garland, Ellen C, Jason Gedamke, Melinda L Rekdahl, Michael J Noad, Claire Garrigue & Nick Gales. 2013. Humpback Whale Song on the Southern Ocean Feeding Grounds: Implications for Cultural Transmission. *PLoS ONE* 8.11, e79422.

Garland, Ellen C, Anne W Goldizen, Melinda L Rekdahl, Rochelle Constantine, Claire Garrigue, Nan

Daeschler Hauser, . . . Michael J Noad. 2011. Dynamic Horizontal Cultural Transmission of Humpback Whale Song at the Ocean Basin Scale. *Current Biology* 21.8, 687-91.

Noad, Michael J, Douglas Cato, H, M M Bryden, Micheline N Jenner & K Curt S Jenner. 2000. Cultural revolution in whale songs. *Nature* 408.537, 537.

Whiten, Andrew, Victoria Horner & Frans B de Waal. 2005. Conformity to cultural norms of tool use in chimpanzees. *Nature* 437, 737-40.

Reviewer #4 (Remarks to the Author):

This is a nice paper, examining the effects of different modes of cultural transmission on the frequency distribution of song types in populations of swamp sparrows. I find the motivation, methods and results generally quite convincing, and it is a good exemplar of the statistical methodology that it uses, as well as interesting in its own right. The implementation of ABC looks fine to me. Many authors have found that ABC model choice generally works well (mindful of paper cited in line 389), as long as the different models give rise to distinguishable prior predictive distributions of summary statistics (as that same set of authors later showed). I do have some (I hope, relatively minor) technical quibbles below.

1) Although not couched in population genetic terms, the basic model has features that could be described as population genetic - copying, transmission, mutation, etc. In that case, I have a worry about the individual identifiability of parameters in the likelihood. The summary statistics are based on the frequency distribution of variant types, as would be the case for the frequency distribution of allele types in a genetic model. However in a genetic model, the approach to equilibrium, and the frequency distribution of variant types is strongly dependent on the population size. Everything is scaled - $N\mu$, Ns , T/N , etc - and usually one works with the scaled parameters to avoid identifiability issues. The authors appear to have separated out the parameters for inference. The magic of Bayesian inference, of course, means that even if parameters are not individually identifiable in the likelihood (as is the case for the standard popgen set-up), by putting a prior on everything you can still get posteriors for all parameters. But this would suggest that the joint posteriors for some sets of parameters are highly correlated, and the marginals are highly dependent on the priors. The authors need to show that this is not a concern here.

2) Related to the above, the approach to equilibrium will be dependent on N (usually in popgen one works with the scaled parameter T/N). However, they choose 5000 generations. To what extent is the choice of 5000 constraining the model?

3) Again, related to above, the abstract comes up with a point value of 1.85%. This is analagous to coming up with a mutation rate from frequency distribution data in population genetics. But to do this you need an estimate of the effective population size. Again, if you report a marginal for one of two parameters that are not individually identifiable, this will be completely dependent on the priors.

Line-by-Line Response to Reviewers' comments:

Reviewer #1 (Remarks to the Author):

First, I think that the claim that this is the first demonstration of conformist bias in natural animal behavior is perhaps overstated and can be given a little more nuance. There are more references to be cited around lines 42-45, particularly in the primate literature...

We agree with this point and have both refined our argument and included the suggested and additional references (lines 43-51). Previous authors have argued that conformist biases (allied to precise learning) are required to maintain stable natural traditions. Our study system and our approach allow us to examine this proposition. In other systems, artificial experimental set-ups are typically required to examine cultural transmission behaviour, which limits their ability to make inferences to large populations and thus make quantitative estimates of the parameters of cultural transmission and the stability of traditions.

Also, a better citation than the review by Morgan and Laland might be [Pike, T. W., & Laland, K. N. (2010). Conformist learning in nine-spined sticklebacks' foraging decisions. *Biology letters*, rsbl20091014.] since the Morgan and Laland review states "The only study of which we are aware that provides clear evidence of non-human animals exhibiting a disproportionate tendency to adopt the majority behavior is Pike and Laland's (2010) investigation of public information use in sticklebacks. Given the taxonomic distance between fish and humans, this finding is most likely to reflect convergent selection for conformity rather than a homologous capability (Laland et al., (2011b)."

We agree and have included this citation in the rewritten introduction (line 46).

Also, the cited Aplin et al. paper is indeed an experiment, but it is one in which a conformist tradition arose naturally in the cultural transmission of an experimental trait (initial association of a color with food availability in a bird feeder led to a transmitted color preference that persisted even when both colors were later associated with food).

Our point here is that because this study focuses on learning an experimental task (in which the initial innovation required taking founders into captivity), it is difficult to generalise from it about the stability of traditions. We have clarified this in our re-written Introduction (line 48-49).

Similarly, in the discussion, the last sentence ["Our findings, then, suggest that the ability to transmit traditions with precision can no longer be considered a fundamental difference between how human and non-human cultures evolve."] might be overstated. The authors don't really make a convincing case that researchers in the relevant fields believe that precise transmission is unique to humans.

The idea that complex culture depends on mechanisms like imitation and teaching because these mechanisms allow precise transmission seems, to us, quite a widely shared one within the "animal cultures debate". We cited earlier in the manuscript the influential paper by Boyd & Richerson and a more recent one by Claidiere. We have added additional references to support our case, including highly cited papers by Tomasello et al (1993; >2000 citations) and Galef (1992; >400 citations). We have added these in both the Introduction (Lines 26-29) and in the final paragraph of the Discussion (Lines 277-279).

Also, I liked that the authors proposed a mechanism for the conformist bias: if birds learn more syllables than they need in a frequency-dependent way and then are more likely to prune rarely heard syllables from their repertoire, an over-representation of frequent syllables might follow. Is there a meaningful distinction to be made, though, between conformist bias in humans, in which conformity might be the actual goal, and this compounding of multiple frequency-dependent processes in birds? I'm asking genuinely; I'm not sure.

This is a good question. On the basis of previous studies, we would argue that conformity may indeed be central to one of the communicative functions of song. We have found that swamp sparrows prefer well-learned versions of songs that they are familiar with (Lachlan et al. 2014, *Proc. Roy. Soc. B*), and conformity would be an obvious strategy to maximise one's performance in such a situation. We argue that the putative strategy for how swamp sparrows achieve conformity might be a simple way to achieve

this strategy. If we are correct, then, as you argue is the case for humans, conformity might be the goal for swamp sparrows. We have rephrased our Discussion to highlight this point (Lines 275-277).

Other things:

I think the paper could better conceptually explain the difference between tutor attractiveness and song attractiveness. Similarly, I conceptually understand model bias, conformist bias, and content bias, but I wanted an explicit comparison of the three and how to distinguish them in practice, especially since they seem like they might be correlated with one another and also occur in combination.

We have rewritten several sections to address this comment. First of all, we have renamed “model bias” as “demonstrator bias” – to avoid confusion between different senses of the word “model” (see Reviewer #1/s comment below). Second, we have edited the paragraph in the Introduction (Lines 91-117) where we introduce the different types of bias to (a) explain the conceptual difference between demonstrator and content biases more clearly, and (b) to much more explicitly explain how each type of bias might influence the frequency distribution of types. Third, we have edited the paragraph in the Methods where we describe how “tutor attractiveness scores” (now “demonstrator attractiveness scores”) are calculated (Lines 373-379).

Could territory size affect a juvenile’s exposure to song? If he grows up on a large territory, is he less likely to hear other conspecifics?

This may be a factor for many songbirds, but not for swamp sparrows in these populations. Swamp sparrows normally (and certainly in the populations we sampled) breed at high density (>150 per km²) with small territory sizes and sing songs that carry across many territories. We have added this point to our description of the populations (Lines 121-122).

I am not sure about the analysis of geographic variation. The methods section does not say what statistics the authors would calculate to accept or reject their hypothesis. It seems like one could do a Mantel test or a spatial autocorrelation analysis given the pairwise values. Based on Figure 2 and the text, the analysis seems qualitative and not quantitative, so I was unsure of the evidence in question when the authors said there was no evidence of preferential syllable sharing with neighbors. Also, the use of kilometers in panel A and meters in panel B made me not immediately realize that the left side of panel A was probably an average of panel B. How large is a typical swamp sparrow territory?

We have added Mantel tests as suggested (Lines 128-133). The diameter of a typical swamp sparrow territory is approximately 50m – information that we have included in the text when we describe the populations. We have relabelled Fig. 2b (now 3b) in kilometres, as suggested.

Terminology: Some of the song terminology is presented in a confusing way. For example, syllable type and song type appear to be used interchangeably (see Figure 1 caption vs. axis labels, but it also occurs in the text), which is perhaps a shorthand that can apply to swamp sparrows, but it does not scale up to many other species and so could be confusing to those who are familiar with other singing styles. In addition, the words “exemplar” and “element” should be defined in the context of this paper.

Could the spectrograms of one swamp sparrow’s full repertoire be included and labeled with the relevant terms?

We apologise for the confusing terminology, which we have rectified along the lines suggested. In swamp sparrows, song learning and syllable learning are nearly synonymous. But in our revised version, we have separated the general process of “song learning” from the specific one of syllable learning in swamp sparrows. We have also modified terminology related to “Model biases”, which is potentially confusing, and have replaced “models” and “tutors” (when used as synonyms) by “demonstrators”. We have qualified what we mean by “exemplar” when we first introduce the word. We have introduced a figure with more explanation of swamp sparrow song structure, as requested (Fig. 1).

In this type of ABC model, do timesteps in the simulation always represent years? The authors could include a few sentences to justify the assumption that one timestep in the model represents a calendar year.

In our simulations, timesteps represent years, but this is not a constraint of the type of model or analysis. For bird song learning, years are appropriate and convenient timesteps, given that swamp sparrows typically only produce one brood per year, and, for juveniles, the process of song learning is limited to their first year of life. We have now explained this in the text (Lines 362-364).

Figure 1. What would the authors like the reader to take away from panels B and C? Just a qualitative sense that the real data look middle-of-the-road and probably most like conformist bias?

From Fig. 1B (now Fig. 2A): the point is that multiple different factors can influence the frequency distributions of syllable types. For Fig. 1C (now Fig. 2B): the primary point is first that there was a high degree of similarity between the six empirical distributions, and that they do indeed most closely resemble conformist bias. We have clarified these points in the Figure Legend.

Figure S1. The peak used to determine the chosen k is not very sharp, with 125 to 200 being pretty similar in GSI. Is this typical for this type of analysis?

Yes – for this type of analysis, silhouette indices are typically not very sharp. Once you have a large number of categories, splitting any one of them is unlikely to have a large effect on the overall cluster validity.

Figure S2. Could you show the prior distribution as well?

We have added the prior distributions to Fig. S2 (now Fig. S3) – note that all were scaled to show a uniform prior distribution.

Minor issues:

Line 34: More than two citations might be good to illustrate the depth of study on conformist transmission.

See above – we have included more references as suggested (Lines 43-48)

Line 55: approximately half of the syllable types are similar to tutor songs; where does the other half come from?

In this lab experiment, the other half appeared to be invented. We have clarified this in our introduction (Lines 63-67)

Line 90: The previous studies mentioned in this sentence should be cited.

We have referred readers to our Supplemental text, where we explicitly re-analyse an earlier study to examine this question (Line 101).

Line 94: “the presence or otherwise of conformist biases” – does otherwise here just mean absence? Or does it imply different degrees?

We have rewritten this sentence (Line 115).

Line 109-112: This is reminiscent of a site frequency spectrum in genetics, and much work has been done to quantify the effects of different forms of selection on these distributions. It might provide an interesting point of comparison.

Indeed this is a good analogy. We hesitate to include it, however, or to draw too heavily on previous work using it, because of the crucial differences between cultural transmission and genetic transmission. Of these, perhaps the most important is that for cultural transmission, we model an individual's repertoire as having several ‘slots’ (and that number varies between individuals), and a given type can only fill one of the slots. For (diploid) genetic transmission, there is a fixed number of two slots, and the same allele can fill both slots (homozygosity).

Line 120 and elsewhere: I found it confusing to have the tutor's song called “model” in the context of the simulation model. For example, I initially read “model biases” to mean “biases in the [simulation] model,” but that is not what the authors mean.

This is a good point. We drew upon Henrich & Boyd's terminology here, in order to try to draw links with related studies. But in the interest of clarity, we have renamed it “demonstrator bias” and have called it this throughout.

Line 129: should this say Fig 1b?

This is now amended (but is now Fig. 2a)

Line 159: “date at which each syllable type was innovated” – how do the authors account for the initial syllables present, or for the lifespan of syllables that are still present at the end of the simulation?

Syllables present at the initiation of the study are given the age 5000 years, since that is how many years the simulation proceeds for. Fig.3 illustrates that this occurred infrequently.

The study looks at a snapshot of the distribution of syllable ages at the end of the simulation (the “current day”). In that sense, each of the syllables types may persist for some period into the future. The analysis allows us to interpret the ages of syllables we might encounter in a cross-sectional study today.

We have edited this sentence for clarity and to explain these points (Lines 203-206)

Line 196: I'm not sure what the authors mean by counterbalance here.

If more syllables are learned in the second phase of learning, the role of the first phase in the decision of which syllable type to learn becomes relatively weaker. W is a parameter that directly determines the weight of the first phase in learning. The positive relationship between W and $NT2$ suggests that the relative role of the two phases is relatively constant across all the accepted simulation runs. We have rewritten the section and added a sentence to clarify this point (Lines 247-249).

Line 203: Does “our ABC-fitted model” refer to Model 1 or Model 2 or both?

Both, and we have clarified by changing model to models (Line 256).

Line 217: Need citations for human culture sentence.

We have added citations here, as requested (Line 279).

Line 267: are the results of this human visual assessment available?

We have included cluster comparison statistics in the text (Lines 327-330).

Line 306: How is alpha chosen? Does it matter how much bigger (or smaller) it is than 1 for the results, and if so, was it estimated from the data or randomly assigned?

Alpha is estimated from the ABC process. That is, it has an initial prior distribution (see Fig. S2C), and the ABC process estimates a range of values for alpha that are consistent with the empirical data, given the simulation. The larger alpha is above 1, the stronger the conformist bias.

We make clear in the text that alpha is one of the parameters estimated by the ABC process (Lines 156-160) and also clarify that alpha is a continuous scale (Line 372-373).

Line 383: “The overall dissimilarity between the simulated and empirical PLS dimensions was calculated as:” it seems like an equation is missing after the colon.

We have added the missing information (Lines 452-454).

Reviewer #2 (Remarks to the Author):

I would highlight that in doing so, the authors need to be more careful – in particular there are some sections of the introduction and discussion that need more careful consideration of existing literature and controversies. I discuss this further below.

We have extensively edited and rewritten our Introduction and Discussion and added additional references, as detailed above and below.

Abstract – “Our results demonstrate conformist bias for the first time in natural animal behaviour...” This is over-reaching. For example, studies of stone-tool use in chimpanzees have claimed to find evidence for conformist bias in the way in which females change their techniques after immigrating into differently behaving groups. Also see evidence in sperm whales (discussed below). I would just shorten this statement to “Our results demonstrate conformist bias in song learning and show that this, ...”

We agree with this suggestion, and have amended the Abstract accordingly (Line 22).

Introduction – The first two paragraphs of the introduction are a confusing mix of the underlying conditions for cumulative culture and for stable traditions. The current literature argues that cumulative cultural evolution (CEE)

requires high fidelity copying (e.g. imitation) and progressive innovations that incorporated into traditions (e.g. through pay-off biased learning). Stable traditions may require relatively 'error-free' precise copying (although not necessarily), and may be facilitated by conformity.

In these two paragraphs, the authors state that CCE is possibly not observed in other animals because traditions are not stable enough, and that humans may also be unique in using social learning strategies. The second of these two statements is demonstrably false – multiple other species have been shown to use social learning strategies (although pay-off biased and conformist strategies may possibly be more restricted). The first, as stated above, conflates the conditions thought necessary for CCE and stable traditions.

I suggest the authors reword these two paragraphs and focus on stable traditions rather than CCE), and make it clear that they are referring to imitation (high fidelity copying) and conformity (making traditions resistant to erosion from errors or from immigration).

While we agree that our Introduction was not as clear as it should have been, we think that our underlying argument is an established one. Tomasello (to whom we now refer) introduced the idea (e.g. Tomasello et al. 1993) that precise learning was necessary for stable traditions, and that stable traditions were required to accumulate culture. A quote from his paper with Tennie et al: “The claim in the original paper was that while inventiveness is fairly widespread among primates, humans transmit cultural items across generations much more faithfully, and it is this faithful transmission (the ratchet) that explains why human culture accumulates modifications over time in a way that chimpanzee and other animal cultures do not.” Boyd & Richerson (1996) also reached a very similar argument, and to us, these arguments seem to have been very influential in the field (see also the comments of Reviewer 3). We certainly do not wish to focus overmuch on cumulative culture, but we do want to explain why there has been a general interest in the stability of animal traditions.

Regarding the second of the points made in this comment, we agree with the reviewer that there is much evidence for transmission biases in animal social learning, and see that our sentence on this was poorly worded. In fact, we intended to make the point that the Reviewer makes parenthetically: there have been arguments that certain types of bias (especially conformity biases) are uniquely human and underlie stable cultural traditions.

We have rewritten this paragraph to make these arguments clearer (Lines, 26-32), including adding a wider range of references.

L34 – I suggest changing “strategies for” to “biases in”, to make the causation a bit less explicit.

We have replaced the text with “social learning strategies” rather than “strategies for social learning” (Line 35), which makes a link with the previous paragraph without such an explicit inference of causation.

L43 – I would also reference van de Waal et al. (2013) Science here. This paper found some evidence for conformity in vervet monkeys.

We have included this (and some additional) references here, as suggested (Lines 44-50).

L44-45 – Conformity in natural behaviour has arguably been shown in primates. Additionally, Cantor et al. (2015) Nature Communications, used a similar simulation based approach in sperm whales, where they modeled data with various biases to produce patterns that they compared with real recorded vocal dialects in whales. They also found that the observed patterns were likely to have arisen through conformist learning (along with homophily).

We have also included a reference to this interesting paper, as suggested (Line 46).

L55 – I'm interested in where the other half of the dialect comes from? Are these an innate syllable set, or innovated by each juvenile?

As far as we can tell with current methods, the syllables were “invented” (to use the authors’ term). Whether or not these invented syllables tended to follow common patterns suggesting an underlying innate syllable set was not (to our knowledge) examined. We now explain that these syllables were invented in our Introduction.

We would like to stress that the point where half of the syllables were “invented” is only at an early stage in development. In the final crystallised song repertoire, the proportion of invented syllables was much lower (we also now explain this in the Intro, Lines 61-65).

L61 – reference?

We have added a reference to the Acerbi et al. paper (see below) that explicitly talks about sampling in this context (Line 72).

L78 – “it is thus particularly suited for studies of cultural transmission” ABC is used for a range of applications; can you justify this statement more explicitly? Also, be aware that there is some current controversy about whether population-level data can be used to look for conformity (Acerbi et al. scientific reports, and others, but see Smaldino et al. Bioxiv). I personally think the approach taken here is fine, but the authors should be clearly cognizant that this argument exists.

We have included a citation to the Acerbi paper (Line 105), which makes almost exactly the same point we do about how content and model/demonstrator biases can mimic the effects of conformist biases. We find that this mimicry is only partial in our case.

L85 – This is not exactly what people will think of as a classic ‘model bias’. I would also include an example more familiar to the readers. For example, ‘copy related individuals’.

We have added such an example – but have chosen “successful individuals that are paired” as something that we think might be both relevant to swamp sparrows, while still serving the purpose of providing a more familiar example (Lines 97-98).

L95 – This is a funny way to frame it the rest of the paper. I would instead just state that you are investigating whether they show evidence of biased transmission, and if so, identify what biases.

We have rewritten this sentence, but have kept a focus on conformity (Lines 114-117), along the lines suggested.

Results – The figures aren’t referred to in order, and I found this very confusing. Can you please reorder them in a more logical fashion? I also suggest a new figure in the main text that summarizes the data collection; showing the six populations and the geographic distance between the samples collected within and between populations. This could be combined in a composite figure with figure 2, which surely should be presented first (and perhaps also with the syllable examples in figure 1a).

We have included a new Fig. 1, as suggested, which introduces the song organisation and syllable variation found in swamp sparrows.

We have also added a new Fig. S1 which presents maps of the populations and territories. We do not believe there will be space to include it in the main document (but are happy to revise this in light of any advice provided by the editor).

We have included a clearer reference to Fig. S4. Other figures were referred to in order, except for Fig. 2b, which was referred to after Fig. 3, but which naturally belongs with Fig. 2a. We think that our ordering is correct, but are very happy to take advice from the editor on this matter.

L140 – In light of other studies showing evidence for preferences for species or subspecies song, it is surprising there is no evidence for content biases. Worth going into more depth on this result, perhaps in the discussion?

Songs that lie outside species/subspecies limit are likely to be highly selected against. We certainly do not rule out such a bias. What we can rule out is that 80% or more of innovations lie outside species norms. When only a small or moderate proportion of innovations are selected against, then their effect on the frequency distribution of songs is not large enough to explain our results.

We have added some material to address this point in the Results section (Lines 182-187).

L145-150 – I find the link between sampling extent and strength of conformity unsurprising, as these two would work together? That is, there is less error in a larger sample, so even a weaker bias will move the population faster towards uniformity. However I agree it is important to highlight this.

Yes, this is exactly how we envisage this relationship, and have used the phrasing of less error with larger samples to drive home this point (Lines 191-193).

L163 – I would qualify this statement a little more, as you don't show this directly. Even just stating "Thus the results of our analysis suggest that ... is potentially capable".

We have qualified the statement as suggested (Lines 209-210).

Discussion – What is the proposed experiment? What should be tested next? I think it would add a lot to your discussion if you detailed what the next steps would be in validating or extending these findings.

This is a good suggestion, which we have implemented. We have highlighted that the predictions of Model 2 should be tested experimentally (Lines 259-261), and that our methods can now be applied to other species to gather together properly comparative data for the evolution of song learning behaviour (Lines 265-269).

L208 – This is an overstatement of your findings. You don't have direct evidence of the past conditions of the populations, only the current distribution of song-types. I don't think your evidence is stronger than, for example, primate archeology that has found stone-tools dating back 1000s of years, or even the New Zealand studies on song of European songbirds.

We have revised as suggested (Lines 261-264).

L222 – "The complexity of song sparrow vocal culture of course does not being to approach that of humans". This qualification isn't necessary, of course it is the case. I don't think the work benefits much from these sorts of direct comparisons with humans. I would instead focus on animal culture, and this emerging field (including the exciting evolutionary and ecological implications of animal culture).

We now clarify that we are directly responding to the debate introduced by Tomasello et al. (Lines 277-279)

L224- again, I don't think it benefits from trying to compare human and non-human culture. This feels like an artificial insertion – imitation is well known in bird song, and you don't discuss imitation much previous to this. Why don't you instead focus on conformity, or on how these results suggest that populations can maintain deep learning-based divisions in behaviour, with evolutionary/ecological implications?

We would suggest that much of the initial interest in these topics stemmed from arguments about the uniqueness of human culture. In our revised Discussion, we now focus more on the significance of our results for bird song research, but we would also like to keep this link to a broader discussion in the social learning / animal culture field.

L238 – did you consider the relative frequency of song types within each male's complete repertoire?

No. As yet, we have no evidence in this species for consistent variations in the frequencies of types within the repertoire.

Reviewer #3 (Remarks to the Author):

1. The biggest issue with the acceptance of any modelling study, regardless of how much data underlies the model, is always the issue of its relationship to reality; this study will be no exception. Taking the viewpoint of a sceptic (and there will be many among the non-biologist target groups listed above), one can always argue that the conclusions follow from the parameters chosen for the model, or that the data do not constrain the model enough.

I think the authors would be able strengthen their message if they can offer some independent indication that

swamp sparrow traditions are indeed long-lived. For example, are there old-recordings from fifty or more years ago, that are properly classified in this cluster hierarchy? (or old spectrograms my old copy of "Birds of North America" from 1966 has spectrograms strikingly similar to those in this paper). I'm not asking for a detailed analysis, but more of an "eyeball what data is available" approach in the Discussion.

We would like to thank the reviewer for this suggestion. It led us to visually compared our recordings with spectrograms published in Marler & Pickert (1984), which represent a complete survey of one marsh in the Hudson Valley population, made in the late 1970's. We found that the commonest type now in 2009 also the commonest type then, and that nearly all the types sung by more than one male in 2009 were present in the 1970's. We have included this information in the Results section of our manuscript (Lines 214-219).

It might also be mentioned that even for the sounds of language, we have only writing to push us back to 500 years, and we just assume that this reflects sound: but detailed speech analysis of vocal dialectal details change within a single lifetime (see Harrington et al 2000)

Although certainly an interesting topic and link to our results, in trying to balance the suggestions of Reviewer 2 (who would prefer less comparison with human culture) and Reviewer 3 here, and in trying to keep the word limit down, we have decided not to follow this suggestion.

Alternatively or in addition, some measure of the fidelity of learning from lab studies with tape tutored animals would help strengthen this core argument. This could simple reference the large literature for this species, most of it by Peter Marler et al.

We are not convinced that a direct comparison with the fidelity of learning in lab studies is advisable. The manipulations involved in hand-rearing, from removing any social context from learning, to using playback through speakers often in conditions with high levels of reverberation, to providing the right diversity and quantity of song playback all likely interfere to some extent with song learning. We have described the evidence from laboratory studies in more detail on Lines 61-67.

2. I think a few more words about the approach should be placed in the main part of the paper (not just the methods). Crucially, it is implied (e.g. line 156) but never stated that the models underlying these simulations are run over many generations, but neither this nor the number of generations simulated is specified. Nor is the mapping from generations to years (though I guess the assumption is that this is one to one would be valid for this seasonally breeding case).

Also, perhaps a sentence about the main differences between the ABC approach and the more standard Bayesian methods (like MCMC-based model testing) would be welcome, and a specification of the advantage(s) of this new approach.

This is a good point. We did aim to provide exactly that kind of introduction to our methods for the reader, but have incorporated these suggestions (e.g. Lines 139-140, and Lines 362-364 for more detail about the use of years as time-steps).

The main advantage of the ABC approach is that it can be applied when likelihood cannot be calculated – such as is the case with our individual-based simulations. We make this point when we introduce ABC in the introduction (Lines 86-90).

3. I don't think the authors try hard enough to connect this work to the long-running "animal culture" controversy (perhaps by design, given the controversy!). A lot of this work focusses on non-vocal social learning (e.g. primates learning tool use from each other in the work of Whiten and colleagues) and shows poor fidelity and rapid decay; this has led many technology-focsed theorists like Boyd and Richerson to the conclusion that high-fidelity, lasting traditions and a cultural ratchet effect are uniquely human.

But this literature often ignores the work on vocal traditions, both in birds (which are well-covered in the current reference list) and in humpback whales, where the work of Noad and Garland could be cited.

We have added more references in the introduction to the work by Whiten as well as Tomosello et al. (Lines 26-29). We have included references of humpback whales (Line 46).

4. Minor typos/odd words:

line 204: some of the mutation rate -> some component of the mutation rate

We accept this point, but have used “some fraction of...” instead (Line 173)

209-210: it can no longer be considered... : the phrasing here is quite awkward and roundabout – please rephrase.

We have rephrased this sentence (Lines 261-264).

Line 534: aspects to individual learning: aspects of...

We have replaced this as suggested (Line 651).

Reviewer #4 (Remarks to the Author):

1) Although not couched in population genetic terms, the basic model has features that could be described as population genetic - copying, transmission, mutation, etc. In that case, I have a worry about the individual identifiability of parameters in the likelihood. The summary statistics are based on the frequency distribution of variant types, as would be the case for the frequency distribution of allele types in a genetic model. However in a genetic model, the approach to equilibrium, and the frequency distribution of variant types is strongly dependent on the population size. Everything is scaled - $N\mu$, Ns , T/N , etc - and usually one works with the scaled parameters to avoid identifiability issues. The authors appear to have separated out the parameters for inference. The magic of Bayesian inference, of course, means that even if parameters are not individually identifiable in the likelihood (as is the case for the standard popgen set-up), by putting a prior on everything you can still get posteriors for all parameters. But this would suggest that the joint posteriors for some sets of parameters are highly correlated, and the marginals are highly dependent on the priors. The authors need to show that this is not a concern here.

We fully agree with the reviewer that identifiability is a major concern in models of our type. As the reviewer, highlights, the classic parameter trade-offs in population genetics involve population size, and we return to that below. To address the issue of identifiability more broadly, we examined correlations between all key parameters' posterior distributions in Table S3, and ensured that we discussed all the larger correlations (i.e. $|r| > 0.4$) in the text (Lines 179-182; 189-192). This approach raised the question of identifiability between conformist bias (α) and the number of demonstrators sampled (NT), and between bird song mutation rate and the proportion of attractive songs (Patt). It is clear from the edited figure S4 that in both the case of NT and Patt, the posterior distributions clearly identify upper and lower (respectively) limits to the parameter values within the broader prior support (Fig. S4 shows both prior and posterior distributions). We hope these plots provide adequate evidence to the reader that our results are driven by the data, and not prior assumptions.

Returning to the issue of population size in particular, we believe that there are two different responses to the question of identifiability with respect to population size and mutation rate. First, as the reviewer suggests, our results do depend to a degree on the priors that we set for population size (log-uniform with limits 400 and 3000). To our best effort, we set priors for population size that were based on (though broader than) estimates of population size for each population. We would like to point out that we have reasonably good estimates of population size in this study. Because of the fact that swamp sparrows prefer a habitat that happens to be very patchily distributed, it is generally relatively straightforward to establish the area of local habitat (Table S1). And our extensive sampling of the population helped establish realistic estimates of population density. We are therefore confident that the population size of male swamp sparrows was >400 for all populations. Setting upper limits for population size is complicated by the fact that we do not have data for the level of migration into and out of populations (although the fact that our results are consistent across populations that vary in their connectedness to other populations might suggest this is relatively rare). To address this, we now mention Table S3 more clearly in the paper and discuss the issue of correlations between mutation rate and population size in depth in the Results (Lines 169-175).

Despite this, we actually found only weak correlation (in the expected direction) between posterior distributions of mutation rate and average population size ($r = -0.201$, Table S3). Our second response is therefore to examine the question of why that might be. We believe that there may be fundamental differences between cultural evolutionary and population genetic models that may partly mitigate this trade-off. In particular, we find that conformist bias appears to break the trade-off between mutation rate and population size in setting population-level patterns of diversity. To demonstrate this we ran two further sets of simulations, drawing 1000 samples from the priors of mutation rate and population size,

varying conformist bias, and keeping all other parameters constant (at the median values found by our ABC analysis). We found that without conformist bias, we generated the predicted trade-off between population size and mutation rate. But when we included conformist bias (setting $\alpha=1.3$), there was no longer such a trade-off. To address this, we have added an additional figure (Fig. S7) and a description of the analysis (Supplemental Text, also Lines 168-169).

2) Related to the above, the approach to equilibrium will be dependent on N (usually in popgen one works with the scaled parameter T/N). However, they choose 5000 generations. To what extent is the choice of 5000 constraining the model?

In pilot analyses, we ran the analyses with 2000 years and found indistinguishable results. In Fig. 4, one can see that very few syllable types even survive for 5000 years in our posterior distributions.

3) Again, related to above, the abstract comes up with a point value of 1.85%. This is analogous to coming up with a mutation rate from frequency distribution data in population genetics. But to do this you need an estimate of the effective population size. Again, if you report a marginal for one of two parameters that are not individually identifiable, this will be completely dependent on the priors.

Please see our discussion in response to point (1) above.

REVIEWERS' COMMENTS:

Reviewer #1 (Remarks to the Author):

The authors have addressed my concerns in this revised manuscript. A few remaining questions:

The Figure 2 caption states "Frequency distributions of syllables types... suggesting a match between the empirical data and the simulations with Conformist Bias" – is there anything more quantitative that can be said here? (also typo in syllable[s] types)

Lines ~110-115 I was a little confused by these sentences:

"Conformist biases increase the frequency of common variants. But if individuals only sample a small proportion of the population, intermediate and rare variants are likely to all be only sampled only once by any one individual, and thus rare variants are not. All three biases might exist simultaneously, and this our simulations took all three into account."

Particularly the phrase "and thus rare variants are not." Also I think "this our simulations" should be "thus our simulations."

Reviewer #2 (Remarks to the Author):

I reviewed this manuscript previously, when I thought it showed some exciting results with an impressive dataset and novel analyses. However it needed some revision of the language to tone-down some of the statements, and some further clarity in the descriptions of analyses and figures. I'm happy with all the revision that the authors have made in response to my comments and the comments by other reviewers. The manuscript reads much better than it did previously, and I like the extra interested added with the comparison to historic recordings of swamp sparrow songs. I only have a few minor comments, and I can recommend the manuscript to be accepted for publication. Great work!

Minor Comments:

L21 – a little bit of over-statement, as the evidence is based on a model. Replace "can" with "could".

L40 – could you give a reference for this sentence? (definition of conformist learning)

L51 – this again is a bit of an overstatement that misrepresents current literature. Could you change "naturally occurring behaviour" or "avian song".

L75 – not necessarily, I could imagine there could be alternative mechanisms that could produce this pattern. What would the copying error rates that have been estimated in other studies?

L115-117 – you could add one more sentence here to give further explanation? The previous paragraph makes it sounds like all biases produce a similar result. You reference the figure, but maybe also the text could make it clearer how the observations are able to differentiate between all of them?

L135 – comma after population

L195 – Could cite Aplin et al. 2017 PNAS here, which also estimated a moderate conformist bias that had a strong population level effect.

L261 – (at the end of the sentence) could add something like “and eventual outcomes”.

L265-8 – Can you add a sentence more to this paragraph to explain further? This seems intriguing, but how would it allow you to test conformist learning conditions? What is the hypothesis? (unrelated, I wonder if Humpback whale song would be a interesting one to apply your model to).

L286 – at least on its own it can't explain human cumulative culture. But it could still do so through interactions with other effects. For example, precise copying + pay-off biased improvements

Reviewer #3 (Remarks to the Author):

The authors have addressed all of my earlier criticisms successfully, and done a very thorough and thoughtful revision of the ms. I am happy to recommend acceptance, as is.

Reviewer #4 (Remarks to the Author):

I am generally supportive of the study, but just had some little niggles about the model, which the authors have addressed. I have looked through the revised document, the responses, and the supplemental text. The authors address the worries I had about scaling of parameters in what is essentially a population-genetic model by demonstrating convincingly (responses, text in paper and supplement, Fig S7) that it is not a big issue when conformist bias is involved in the model. If my analogical thinking is correct 'conformist bias' corresponds to positive frequency dependent selection. I have to admit I will need to think harder about the problem more generally because it implies some internal calibration - it suggests that it is possible to obtain rates, and therefore timescales directly from frequency data, which is a bit of a puzzle for me. But certainly I do not want to hold up this interesting study on account of my potentially faulty reasoning because the authors' results would suggest it is not an issue.

Response to reviewers.

Reviewer #1 (Remarks to the Author):

The Figure 2 caption states “Frequency distributions of syllables types... suggesting a match between the empirical data and the simulations with Conformist Bias” – is there anything more quantitative that can be said here? (also typo in syllable[s] types)

We have amended the typo. We would like to stress that this aspect of the figure is illustrative. The “more quantitative” comparison between the simulations and the empirical data is exactly what the main ABC analysis is for.

Lines ~110-115 I was a little confused by these sentences:

“Conformist biases increase the frequency of common variants. But if individuals only sample a small proportion of the population, intermediate and rare variants are likely to all be only sampled only once by any one individual, and thus rare variants are not. All three biases might exist simultaneously, and this our simulations took all three into account.”

Particularly the phrase “and thus rare variants are not.” Also I think “this our simulations” should be “thus our simulations.”

We have amended the typo (“thus”), and have rephrased this sentence – which we apologise for leaving in a confusing state.

Reviewer #2 (Remarks to the Author):

Minor Comments:

L21 – a little bit of over-statement, as the evidence is based on a model. Replace “can” with “could”.

We have amended, as suggested

L40 – could you give a reference for this sentence? (definition of conformist learning)

We have added a reference to Henrich & Boyd, 1998

L51 – this again is a bit of an overstatement that misrepresents current literature. Could you change “naturally occurring behaviour” or “avian song”.

We are not sure exactly how this was an overstatement. We have rephrased to avoid any possible implications regarding any other studies.

L75 – not necessarily, I could imagine there could be alternative mechanisms that could produce this pattern. What would the copying error rates that have been estimated in other studies?

There have been few previous attempts to quantify copying error. We cite two examples at this point in the manuscript. No attempts have been made for swamp sparrows before.

We have rephrased this, but think that our point stands – if many birds share songs, it suggests (not proves!) that learning is often precise. The only other logical possibility is that birds reinvent songs already present in the population, and as yet there is no evidence that this can happen (because songs are highly multidimensional, the probability of this happening is quite low).

L115-117 – you could add one more sentence here to give further explanation? The previous paragraph makes it sounds like all biases produce a similar result. You reference the figure, but maybe also the text could make it clearer how the observations are able to differentiate between all of them?

We have rewritten the sentence before (see also response to Reviewer 1 above) to make clearer how conformist biases might be expected to generate slightly different effects on frequency distributions (specifically boosting commoner variants, rather than removing rarer ones).

L135 – comma after population

Amended as suggested

L195 – Could cite Aplin et al. 2017 PNAS here, which also estimated a moderate conformist bias that had a strong population level effect.

This was a good idea, and we have added the reference as suggested.

L261 – (at the end of the sentence) could add something like “and eventual outcomes”.

Amended as suggested

L265-8 – Can you add a sentence more to this paragraph to explain further? This seems intriguing, but how would it allow you to test conformist learning conditions? What is the hypothesis? (unrelated, I wonder if Humpback whale song would be a interesting one to apply your model to).

We have expanded this paragraph as suggested.

L286 – at least on its own it can't explain human cumulative culture. But it could still do so through interactions with other effects. For example, precise copying + pay-off biased improvements

We absolutely agree with this comment, but think that our point still stands.

Reviewer #4 (Remarks to the Author):

I am generally supportive of the study, but just had some little niggles about the model, which the authors have addressed. I have looked through the revised document, the responses, and the supplemental text. The authors address the worries I had about scaling of parameters in what is essentially a population-genetic model by demonstrating convincingly (responses, text in paper and supplement, Fig S7) that it is not a big issue when conformist bias is involved in the model. If my analogical thinking is correct 'conformist bias' corresponds to positive frequency dependent selection. I have to admit I will need to think harder about the problem more generally because it implies some internal calibration - it suggests that it is possible to obtain rates, and therefore timescales directly from frequency data, which is a bit of a puzzle for me. But certainly I do not want to hold up this interesting study on account of my potentially faulty reasoning because the authors' results would suggest it is not an issue.

Conformist bias might be seen as a particular type of positive frequency-dependent (cultural) selection. It relies, however on sampling of small proportions of the population by learners, and this leads to particular a particular type of relationship between frequency and fitness.